# Gender Disparity in Oral Anticoagulation Therapy in Hospitalised Patients with Atrial Fibrillation During the Ongoing Syrian Conflict: Unbalanced Treatment in Turbulent Times

**DOI:** 10.3390/jcm14041173

**Published:** 2025-02-11

**Authors:** Ibrahim Antoun, Alkassem Alkhayer, Aref Jalal Eldin, Alamer Alkhayer, Khaled Yazji, Riyaz Somani, G. André Ng, Mustafa Zakkar

**Affiliations:** 1Department of Cardiovascular Sciences, University of Leicester, Glenfield Hospital, Groby Road, Leicester LE3 9QP, UK; riyaz.somani@uhl-tr.nhs.uk (R.S.); andre.ng@leicester.ac.uk (G.A.N.); 2Faculty of Medicine, University of Aleppo, Aleppo P.O. Box 12212, Syria; 3Department of Internal Medicine, University of Tishreen’s Hospital, Latakia P.O. Box 2230, Syria; alkassemalkhayer92@gmail.com (A.A.); aref.jala.eldin@gmail.com (A.J.E.); alameralkhayer@hotmail.com (A.A.); 4NIHR Leicester Biomedical Research Centre, Leicester LE3 9QP, UK; k.yazji@theviewhospital.com; 5Department of Cardiac Surgery, University Hospitals of Leicester NHS Trust, Glenfield Hospital, Leicester LE3 9QP, UK; 6Department of Cardiology, The View Hospital, Ad-Dauha, Qatar; 7Department of Cardiology, University Hospitals of Leicester NHS Trust, Glenfield Hospital, Leicester LE3 9QP, UK; 8Faculty of Medicine, University of Damascus, Damascus, Syria

**Keywords:** cardiovascular disease, developing world, females, males, direct oral anticoagulant, vitamin K antagonist

## Abstract

**Background:** Disparities in the therapy and outcomes of males and females with atrial fibrillation (AF) are known in the developed world. Still, data regarding these disparities in the developing world are scarce. This study explores gender trends and outcomes in oral anticoagulation prescription during the Syrian conflict. **Methods:** We included adult patients with an index admission with AF to Latakia’s tertiary centre between September 2021 and February 2024. Data regarding treatments and comorbidities were taken from patients’ medical notes. The composite outcome was a readmission with a cerebrovascular event (CVA) or a bleeding event within 60 days of index discharge. A regression model was used to assess predictors of composite outcomes. **Results:** A total of 683 consecutive patients admitted and treated for AF satisfied the study criteria, of whom 347 (51%) were females. In patients with a guideline indication for anticoagulation (*n* = 553), males were prescribed more DOACs and fewer VKAs than females (93% versus 71%, *p* < 0.001 and 7% versus 29%, *p* = 0.01, respectively). Composite outcomes occurred significantly more in females than males (16% versus 6%, *p* = 0.03). Females on VKAs had significantly more composite outcomes than males (70% versus 53%, *p* < 0.001). Independent predictors of composite outcomes included females compared to males (hazard ratio [HR]: 2.3 and 6.2, 95% confidence interval [CI]: 1.3–4.2 and 3.7–10.8, *p* = 0.001 and <0.001) and VKAs compared to direct oral anticoagulants (DOACs) (HR: 8.4, 95%CI: 4.8–15.3, *p* < 0.001). **Conclusions:** Females at this Syrian centre had a higher use of VKAs and a lower use of DOACs compared to males, resulting in a higher rate of composite outcomes of CVA and bleeding events.

## 1. Background

Atrial fibrillation (AF) is the most common sustained arrhythmia globally, significantly impacting morbidity and mortality rates, particularly from stroke and systemic embolism [1]. AF prevalence is growing, especially in middle- to low-income countries, where it is likely underestimated due to limited healthcare resources and infrastructure and under-reporting [2]. The clinical management of AF is critical, given its role in elevating the risk of severe complications. Despite its global burden, there is a striking scarcity of data on AF demographics, management and outcomes in the Middle East, particularly in Arab countries, where AF-related research comprises only 0.7% of the total global AF studies [3]. Only four epidemiological registries exist in this region, reflecting the under-representation of AF-related research [4]. Therefore, understanding AF in this context is essential for developing tailored public health strategies to mitigate its impact.

Vitamin K antagonists (VKAs), such as warfarin, have been the primary choice for stroke prevention in AF for years. According to a meta-analysis, DOACs have been incorporated into clinical practice with superior outcomes for non-valvular AF [5]. However, the safe and effective use of VKAs is resource-intensive. It is challenging as it requires careful dose titration and regular blood tests to maintain the international normalised ratio (INR) within a therapeutic range. When VKAs were the only oral anticoagulation option for stroke prevention, data suggested undertreatment in women compared with men [6], even though the harms of therapy, including major bleeding, were comparable [7]. The introduction of DOACs revolutionised AF management, offering a safer and more convenient alternative with fewer monitoring requirements and superior outcomes in non-valvular AF [8]. However, their use in developing countries remains limited due to price and accessibility [9]. Gender disparities in AF treatment are well documented globally, as highlighted by recent studies and meta-analyses [10,11,12,13]. Studies from developed countries demonstrated that females with AF are less likely to be given anticoagulation compared to males [10,14,15,16]. Furthermore, females were proposed to be under-anticoagulated [17]. Additionally, gender-specific barriers, including healthcare access, physician bias, and patient preferences, have been identified as key contributors to these disparities [18].

The potential reasons for these disparities are multifactorial, including physician bias, differences in risk perception, and socioeconomic factors (8). In contrast, others showed no sex disparity in anticoagulation prescription [19]. Such data are sparse in the context of developing nations, where access to DOACs remains limited due to cost and availability. For example, a meta-analysis highlighted that females in high-income settings benefit significantly from DOACs, experiencing lower rates of major bleeding and ischaemic events compared to VKAs [8]. These findings underscore the importance of equitable access to DOACs in addressing gender disparities in AF outcomes. This can be challenging in developing countries undergoing conflict where data are scarce and the financial situation is challenging. Globally, gender disparities in healthcare are well documented, with women often receiving less aggressive treatment for cardiovascular diseases and facing barriers to accessing advanced medical therapies [20]. These inequities are magnified in crisis settings, where societal disruption, economic instability, and reduced healthcare capacity disproportionately impact women. Gender disparities in AF management are particularly concerning due to the compounding impact of anticoagulation choice and systemic barriers. Females with AF are more likely to experience adverse outcomes with VKAs due to challenges in maintaining therapeutic INR, a situation worsened by resource limitations. Despite the superior safety and efficacy profile of DOACs, these treatments are less accessible to females in many resource-limited settings due to cost and sociocultural barriers. Furthermore, evidence on gender disparities in AF management is scarce in conflict-affected regions like Syria, where healthcare infrastructure and resources have been severely compromised. In Syria, the protracted conflict has exacerbated these challenges, with healthcare infrastructure decimated and economic hardship limiting access to medications like DOACs. Additionally, societal norms and caregiving roles may further restrict women’s ability to seek timely and adequate care, leading to disparities in treatment and outcomes. Understanding how these global trends manifest in conflict settings is critical for addressing gender-based inequities in healthcare delivery [20]. Since 2011, Syria has been embroiled in a protracted conflict that has precipitated one of the most severe humanitarian crises in recent history. The ongoing war has led to the mass displacement of millions and a catastrophic depletion of healthcare resources, exacerbated further by the COVID-19 pandemic and recurring cholera outbreaks [21,22]. Over 50% of the healthcare workforce has fled the country, and less than half of its hospitals are functioning at full capacity, with up to three-quarters of healthcare providers fleeing the country in search of safety [23]. This has placed mental pressure and increased workload on the remaining medical staff [23]. These challenges have severely compromised the delivery of healthcare, including the management of chronic conditions like AF.

Cultural factors, such as societal norms limiting women’s autonomy in healthcare decision making, may delay or reduce access to optimal treatments like DOACs. Economic challenges, exacerbated by the conflict, disproportionately affect women, potentially limiting their ability to afford newer therapies. Additionally, implicit physician bias may contribute to disparities in prescription patterns, with physicians defaulting to VKAs for females based on assumptions about compliance or bleeding risks. Furthermore, systemic challenges within the healthcare system, such as the scarcity of DOACs and the logistical feasibility of INR monitoring for VKAs, likely played a significant role in shaping treatment disparities. AF management in Syrian hospitals remains largely unexplored, with few recent studies providing insights into inpatient management [24,25,26]. However, a critical gap remains in understanding how AF is managed in Syria’s ongoing conflict and resource limitations. For example, a recent study demonstrated that females with AF had a poorer quality of life than males, necessitating a study to explore gender differences in treatments offered [27].

A detailed, real-world analysis of current AF care practices and outcomes is urgently needed to identify deficiencies and propose feasible, context-specific solutions. Moreover, it is essential to explore whether the introduction of DOACs in this conflict-affected setting has had any impact on reducing gender disparities in AF treatment, an area that remains under-researched.

In this study, we aimed to evaluate the trends in oral anticoagulation prescriptions for males and females hospitalised with newly diagnosed AF and compare the effect of gender and anticoagulation prescription on outcomes in a tertiary care centre in Syria.

## 2. Methods

### 2.1. Study Design and Patient Selection

This single-centre retrospective observational cohort study was conducted at Tishreen’s University Hospital, Latakia, Syria, between September 2021 and February 2024. The hospital is a large government-operated public institution associated with Tishreen University, and it serves as the tertiary healthcare centre for the city and the surrounding areas. The hospital has around 860 beds and provides free healthcare. On average, the hospital sees approximately 50,000–60,000 inpatients yearly, with an even more significant number of outpatients seeking care in various medical departments. The Cardiology Department at Tishreen University Hospital comprises around ten general cardiology consultants. The department is well equipped to handle both emergency and elective cases of cardiovascular diseases, and it has access to state-of-the-art cardiac diagnostic tools, including echocardiography and cardiac catheterisation facilities. During the study period, the city was politically controlled by the government without ongoing active war or new territories gained by military forces.

This study included all patients over 18 years old who were admitted to the hospital and treated for new AF as the primary diagnosis for admission. Exclusion criteria included patients with known AF, patients with metallic heart valves, moderate to severe rheumatic mitral valve disease, and patients on haemodialysis because DOACs are contraindicated in this cohort. Due to healthcare limitations and the scarcity of medical equipment, paroxysmal and persistent AF cannot be specified. The AF diagnosis was proposed by the medical consultant or the medical registrar after discussion with the medical consultant. Hospital records were monitored for 60 days for readmissions. This study was designed according to STROBE guidelines (7). The research reported in this article adhered to the Declaration of Helsinki. The project was conducted as part of an audit reviewed and was approved by the hospital board (reference: 277/B) and involved prospective analysis of retrospectively collected anonymised data. Therefore, the hospital board waived the need for consent.

### 2.2. Data Collection and Variables

Data were collected from hospital paper and electronic records, which were used to establish patient demographics and admission details, including the initiated anticoagulant The CHA_2_DS_2_Vasc score was calculated using the demographics collected. The hospital records were also examined for readmissions within 60 days of index discharge. Variables in this study included the following:-AF diagnosis was established by the medical registrar, medical consultant or cardiology consultant after reviewing the 12-lead electrocardiogram.-A bleeding event is defined by a classification of 2 or above on the Bleeding Academic Research Consortium (BARC) scale [28]. The CALIBER group validated the classification system using electronic health records and linked bespoke studies [29].-A cerebrovascular event (CVA) is a neurological deficit caused by an ischaemic event in the central nervous system. After assessing the patient, a medical consultant clinically diagnoses a CVA.

### 2.3. Study Outcomes

The outcomes of interest included the following:The differences in VKA and DOAC prescription between males and females;The composite outcome comparison between males and females, which is defined as readmission within 60 days due to a CVA or bleeding event;Predictors of composite outcomes.

### 2.4. Statistical Analysis

Continuous variables are expressed as mean and standard deviation (SD). Categorical variables are expressed as counts and percentages (%). Pearson’s χ^2^ or Fisher’s exact test was used for categorical variables between groups. Student’s *t*-tests and Kruskal–Wallis tests were used to compare continuous variables between the groups depending on the normality of the distribution. We used Cox regression to examine the relationship between gender, anticoagulants, and composite outcomes. Our multivariable model was constructed a priori, incorporating potential confounders, including gender, type of anticoagulation, and comorbidities associated with composite readmission outcomes (e.g., hypertension, diabetes, cerebrovascular disease, and CHA_2_DS_2_-VASc score). The goodness-of-fit and discriminatory power of the model was assessed using the G-test and C-statistic, respectively, to ensure robustness. Although detailed data on socioeconomic status, access to healthcare, and INR monitoring were unavailable, we aimed to account for healthcare disparities in this conflict-affected setting partially. We assessed the goodness-of-fit and discriminatory power using the G-test and C-statistic, respectively. A G-test with *p* < 0.05 indicates that the model with the covariates provides a significantly better fit than the null model [30]. A C-statistic of 0.7–0.8 indicates an acceptable model discrimination ability, while a C-statistic > 0.8 indicates a strong model discrimination ability [31]. A 2-sided *p*-value < 0.05 was considered statistically significant. Statistical analysis was performed using GraphPad Prism V10.0 for Mac (San Diego, CA, USA).

## 3. Results

### 3.1. Patient Characteristics and Gender Disparity in Anticoagulation Prescription

A total of 683 consecutive patients admitted and treated for AF satisfied the study criteria, of whom 347 (51%) were males and 336 (49%) were females. Table 1 demonstrates that males and females had similar demographics and anticoagulation treatments.

Details on the anticoagulation treatments are shown in Table 2. Oral anticoagulation administration was given to 502 patients (73%), of which 262 were males (76%) and 240 were females (71%), *p* = 0.01. Compared to males, females had fewer DOACs (69% versus 51%, *p* < 0.001) and more VKAs (23% versus 9%, *p* < 0.001). Of 553 (81%) with a guideline indication for anticoagulation, the proportion of oral anticoagulation prescribed was similar between males and females (74% versus 66%, *p* = 0.08). However, males were prescribed more DOACs and fewer VKAs than females (93% versus 71%, *p* < 0.001 and 7% versus 29%, *p* = 0.01, respectively). Of the 130 patients (19%) without a guideline indication for anticoagulation, there was no significant difference in anticoagulation prescribed between males and females.

### 3.2. Composite Outcomes

Composite outcome details are shown in Table 3. After a median follow-up of 62 days (interquartile range: 58–66 days), composite outcomes occurred in 76 patients (11%). Females had more composite outcomes compared to males (57 [17%] versus 19 [6%], *p* = 0.03). Composite outcomes occurred significantly more times in anticoagulated females compared to anticoagulated males (84% versus 68%, *p* = 0.02). Furthermore, females on VKAs had significantly more composite outcomes than males on VKAs (70% versus 53%, *p* < 0.001)

### 3.3. Predictors of Composite Outcomes

Cox regression analysis regarding predictors of composite outcomes is presented in Table 4. Univariable Cox regression showed that females (HR: 5.2, 95% CI: 2.7–9.8, *p* < 0.001) and VKA use compared to DOAC (HR: 10.2, 95% CI: 4.7–37.2, *p* < 0.001) were associated with an increased risk of composite outcomes. Similarly, multivariate analysis showed that females compared to males (HR: 5.6, 95% CI: 2.4–11.2, *p* < 0.001) and VKA use compared to DOAC (HR: 7.6, 95% CI: 2.6–15.7, *p* < 0.001) were independently associated with an increased risk of composite outcomes. The G-test goodness-of-fit was 66, *p* < 0.001, and the C-statistic was 0.81, 95% CI: 0.75–0.86, indicating strong discrimination ability.

## 4. Discussion

This is the first study describing trends of gender differences in anticoagulation prescription and outcomes in newly diagnosed AF in Syria. This study highlights multiple significant novel findings for this Syrian population. First, in patients with a guideline indication for oral anticoagulation, females were prescribed fewer DOACs and more VKAs. Second, composite outcomes were significantly higher in females compared to males and in patients prescribed VKAs compared to DOACs. Third, independent risk factors for composite outcomes included females compared to males and VKA usage compared to DOAC.

As AF is known to have enormous implications on economies worldwide, recent studies have focused on many aspects of AF, including treatment patterns, hospitalisation, and readmission rates [32]. Previously, developed world studies examining the relationship between gender and the prescription of oral anticoagulation therapy have produced conflicting results. Some studies have found no significant difference in the rates of oral anticoagulation prescription between females and males, which agrees with our findings [33]. In contrast, others have indicated that females were less likely to be orally anticoagulated [14,34]. A recent study in Jordan showed that males were less likely to receive oral anticoagulation than females [24]. This finding contrasts with our study conducted in Syria, where no significant difference was found between the genders, as a similar proportion of males and females with a guideline indication for anticoagulation received oral anticoagulant treatments. The variation between these findings could be due to several factors, including differences in patient populations, healthcare practices, and gender-based health perceptions in the two countries. Furthermore, Jordan is a relatively stable Westernised country, unlike Syria, which has been heavily involved in a 13-year-old conflict severely damaging infrastructure, making generalising the data between the two countries challenging. This disparity underscores the importance of considering regional and contextual differences when interpreting medical research results and applying them across populations.

Additionally, the depletion of healthcare resources due to conflict—including shortages of medical supplies, reduced availability of medications, and a diminished workforce—likely created systemic barriers that disproportionately affected vulnerable populations, including women and displaced individuals. These factors emphasise the complex interplay between migration, resource scarcity, and healthcare disparities in conflict settings

In our study, there was a tendency to prescribe more VKAs than DOACs, especially in females. Historical patterns and familiarity with VKAs may have influenced these prescription behaviours. A recent Syrian-based survey demonstrated that only 22–25% understood and practised prescribing DOACs [34]. Therefore, efforts should be sought to educate Syrian physicians regarding safe, evidence-based DOAC prescriptions. VKAs were given significantly more to females than males in our cohort, which is not in keeping with a recent national study in Scotland suggesting that females were less likely to be given VKAs [14].

Differences in prescription practices, clinical guidelines, and provider bias between genders could also contribute to these discrepancies. For instance, recent studies have shown that females are less likely to be prescribed DOACs despite their superior efficacy and safety profile compared to VKAs [10,17,35]. Additionally, cultural and socioeconomic barriers, particularly in resource-constrained settings, exacerbate these disparities, limiting women’s access to advanced therapies [36]. Physicians might have different thresholds or guidelines for prescribing VKAs based on patient gender, influenced by various factors, including clinical judgment or historical practices. Furthermore, whether conscious or unconscious, provider bias may play a role. Providers may perceive women as being at higher risk for bleeding complications or other side effects and may opt for VKAs, which have more easily controlled dosing and a longer track record.

Furthermore, VKAs need regular INR checks. This could be more practical for females who may not experience the same mobility limitations as men engaged in combat. As a result, they might be considered better candidates for VKAs, as regular follow-ups are required. On the other hand, males, especially those caught up in conflict or displaced by war, might have been given DOACs more often because they are easier to use in unstable conditions. Unlike VKAs, DOACs do not require frequent blood testing, making them better suited for patients who cannot regularly access healthcare for monitoring. Addressing these disparities requires a deep understanding of the healthcare landscape in Syria, including economic factors, healthcare infrastructure, provider training, and patient demographics, to gain clarity on gender differences in anticoagulant prescription.

Furthermore, the ongoing Syrian conflict has led to widespread displacement and resource scarcity, both of which may have influenced the characteristics of the patient cohort. Latakia, where this study was conducted, is one of the relatively more stable regions in Syria, attracting internally displaced populations seeking safer living conditions and better healthcare. This migration may have resulted in a heterogeneous patient cohort that includes individuals from different socioeconomic and geographic backgrounds, potentially affecting access to healthcare and anticoagulation therapies. For instance, displaced individuals may face financial constraints, lack of regular follow-up, and limited access to advanced treatments like DOACs, which could contribute to the observed disparities in outcomes.

Head-to-head analysis and regression analysis demonstrated that females and VKAs were associated with an increased risk of 60-day readmission with a bleeding event or CVA (composite outcome). This is not in keeping with developed world data, which suggested that DOAC reduced CVA in males and females while it increased significant bleeding compared to warfarin in females [37]. The results in our cohort could be because VKAs require regular blood tests, which are paid out of pocket by resource-poor citizens. Although no previous research has examined access to INR testing in Syria, we know that access to healthcare has been challenging, according to conflict-based reports [38,39]. The diminished health infrastructure and resource constraints in Syria likely contributed to the observed disparities in outcomes [38]. For example, the limited availability of DOACs and challenges in INR monitoring for VKAs may have disproportionately affected females, who were prescribed VKAs more frequently than males. Although detailed socioeconomic and access-related data were unavailable, our regression model included gender and anticoagulation type to account for these systemic factors.

Furthermore, primary care is severely impaired nationwide, and following up with patients with regular blood tests is challenging, leading to less time in the therapeutic range in VKA patients. Also, medications in Syria are mainly sold over the counter, and self-medication is highly prevalent, according to a recent study [40]. This can lead to potential interaction with VKAs, increasing or decreasing the INR, and leading to CVA or bleeding events. Care is sought at the last minute, and comorbidities are poorly managed, which can contribute to CVA and bleeding risk. These challenges have severely compromised the delivery of healthcare, including the management of chronic conditions like AF. Cultural factors, such as societal norms limiting women’s autonomy in healthcare decision making, may delay or reduce access to optimal treatments like DOACs. Economic challenges, exacerbated by the conflict, disproportionately affect women, potentially limiting their ability to afford newer therapies.

Additionally, implicit physician bias may contribute to disparities in prescription patterns, with physicians defaulting to VKAs for females based on assumptions about compliance or bleeding risks. Furthermore, systemic challenges within the healthcare system, such as the scarcity of DOACs and the logistical feasibility of INR monitoring for VKAs, likely played a significant role in shaping treatment disparities. Understanding how these global trends manifest in conflict settings is critical for addressing gender-based inequities in healthcare delivery. Our study provides actionable insights for policymakers and healthcare leaders. For example, subsidising DOACs or establishing targeted financial support programs for females may reduce disparities in AF management. Furthermore, training programs for healthcare providers in conflict zones should emphasise the importance of gender equity in evidence-based treatment. Finally, strengthening primary care and community outreach programs can improve INR monitoring and medication adherence, thereby mitigating the risks associated with VKAs in resource-constrained environments. Future studies should incorporate qualitative data from physicians and patients to understand these mechanisms and explore perceptions, biases, and barriers influencing treatment decisions. Additionally, capturing data on socioeconomic status, healthcare literacy, and access to healthcare facilities would provide a more comprehensive understanding of the observed disparities. Future work should also incorporate direct measures of healthcare access and resource availability to capture the impact of these variables on treatment outcomes in conflict-affected populations.

## 5. Limitations

Both study arms were roughly matched in demographics except for the oral anticoagulant prescribed and the CHA_2_DS_2_Vasc score, given that the females obtain higher for gender. However, this study did not come without limitations. Data collection was limited to a single tertiary care centre in Latakia. This city was relatively less affected by the Syrian conflict than the other northern and eastern regions of Syria. Additionally, migration patterns may have influenced the cohort, as internally displaced individuals often relocate to safer regions like Latakia. This heterogeneity could introduce biases related to differences in healthcare access, socioeconomic status, and prior medical care, limiting the generalizability of our findings to the broader Syrian population. Future studies should aim to include data from multiple regions to better understand the impact of migration and resource scarcity on healthcare disparities in conflict settings. Routine INR was conducted in the study centre due to resource constraints. Furthermore, this study lacked detailed follow-up data on adherence to anticoagulation therapy and time in the therapeutic range (TTR) for VKAs, which could have provided additional insight into the observed gender disparities in outcomes. This limits the ability to draw definitive conclusions about the mechanisms underlying the observed gender disparities. This limitation highlights the need for longitudinal studies with robust follow-up to explore these factors in greater detail. Future studies should aim to incorporate prospective follow-up data to better understand the impact of adherence and INR monitoring on outcomes. The exact rhythm and rate control medications, their doses, and the antiplatelet doses were unavailable. There was difficulty in distinguishing primary and secondary AF because of the lack of comprehensive diagnostic data, including detailed biochemical evaluations. 

Additionally, our analysis included only routinely collected data within the medical records and by the number of patients who presented to the hospital. Therefore, other variables potentially impacting outcomes may have yet to be identified. As this is a single-centre analysis, readmission to other centres might have been missed, and the rate of ischaemic and bleeding events might have been underestimated. This study did not address other treatments given during admission, which could have affected this study’s outcomes. Medication compliance data were unavailable in this study, which could explain the outcomes.

## 6. Conclusions

In Syrian patients with newly diagnosed AF, receipt of DOACs was substantially lower among females. Females had a higher risk of hospitalisation for ischaemic or bleeding events compared to males, which is attributed to more VKA usage. These results highlight a critical need for Syrian healthcare workers to address gender disparities in AF treatment in Syria to ensure equal and anticoagulation therapy to optimise outcomes.

## Figures and Tables

**Table 1 jcm-14-01173-t001:** Gender-stratified baseline characteristics of Syrian patients admitted to hospital with primary atrial fibrillation.

	Total (683)	Male (*n* = 347)	Female (*n* = 336)	*p*-Value
Cardiovascular risk factors, *n* (%)				
Age (years)	60 ± 11	61 ± 9.5	60 ± 13	0.32
Hypertension	229 (34%)	127 (37%)	102 (30%)	0.09
Ischaemic heart disease	123 (18%)	57 (16%)	66 (20%)	0.32
Diabetes mellitus	151 (22%)	83 (24%)	68 (20%)	0.27
Cerebrovascular event	132 (19%)	72 (21%)	60 (19%)	0.33
Congestive heart failure	141 (21%)	76 (22%)	65 (19%)	0.45
PCI during last year	40 (6%)	17 (5%)	23 (7%)	0.33
CABG during last year	24 (4%)	7 (2%)	14 (4%)	0.12
CHA_2_DS_2_Vasc score *	2.3 ± 1.4	1.8 ± 1.1	2.7 ± 1.6	<0.001
Other comorbidities, *n* (%)				
Anaemia	111 (16%)	63 (18%)	48 (14%)	0.18
Thyroid disease	24 (4%)	8 (2%)	16 (5%)	0.1
Dementia	51 (7%)	28 (8%)	23 (7%)	0.56
Active malignancy	28 (4%)	15 (4%)	13 (4%)	0.85
Chronic liver failure	44 (6%)	19 (5%)	25 (7%)	0.35
Chronic lung disease	75 (11%)	40 (12%)	35 (10%)	0.71
Atrial fibrillation drug therapy, *n* (%)				
Oral antiarrhythmics	327 (48%)	155 (45%)	172 (51%)	0.08
Oral rate control	425 (62%)	203 (59%)	222 (66%)	0.11
Laboratory results, median (IQR)				
Haemoglobin (g/L)	122 (101–146)	124 (114–132)	120 (99–141)	0.7
Total cholesterol (mmol/L)	4.2 (2.4–6.7)	3.9 (2.1–5.4)	4.4 (3.2–5.9)	0.38
Creatinine (micromol/L)	114 (82–149)	107 (92–121)	120 (103–125)	0.1

***** After excluding gender from the CHA_2_DS_2_Vasc score, the score in females dropped to 1.7 ± 1.4, and comparison with males did not yield statistically significant results (*p* = 0.09). PCI: primary coronary intervention. CABG: coronary artery bypass graft. DOAC: direct oral anticoagulation. VKA: vitamin K antagonist.

**Table 2 jcm-14-01173-t002:** Anticoagulation use based on the CHA_2_DS_2_Vasc score in the study cohort.

	Total (683)	Male (*n* = 347)	Female (*n* = 336)	*p*-Value
Overall anticoagulation use				
Oral anticoagulation	502 (73%)	262 (76%)	240 (71%)	0.68
DOAC *	411 (82%)	241 (69%)	170 (51%)	<0.001
VKA *	91 (18%)	21 (6%)	70 (21%)	<0.001
Guideline indication for anticoagulation defined by a CHA_2_DS_2_-VASc score of ≥1 in males and ≥2 in females				
Guideline indication for anticoagulation	553 (81%)	323 (93%)	230 (68%)	<0.001
Oral anticoagulation **	477 (86%)	256 (74%)	221 (66%)	0.08
DOAC **	394 (83%)	237 (93%)	157 (71%)	<0.001
VKA **	83 (17%)	19 (7%)	64 (29%)	<0.001
No guideline indication for anticoagulation defined by a CHA_2_DS_2_-VASc score of 0 in males and <2 in females				
No guideline indication for anticoagulation	130 (19%)	24 (7%)	106 (32%)	<0.001
Oral anticoagulation ***	25 (19%)	6 (25%)	19 (18%)	0.22
DOAC ***	17 (68%)	4 (67%)	13 (68%)	0.89
VKA ***	8 (32%)	2 (33%)	6 (32%)	0.97

DOAC: direct oral anticoagulation. VKA: vitamin K antagonist. * Percentages are calculated from patients on anticoagulation. ** Percentages are calculated in patients with a guideline indication for anticoagulation. *** Percentages are calculated in patients without a guideline indication for anticoagulation.

**Table 3 jcm-14-01173-t003:** Composite outcome details between males and females.

	Total (76)	Male (*n* = 19)	Female (*n* = 57)	*p*-Value
Oral anticoagulation	61 (80%)	13 (68%)	48 (84%)	0.02
DOAC	11 (14%)	3 (16%)	8 (14%)	0.78
VKA	50 (66%)	10 (53%)	40 (70%)	<0.001

VKA: vitamin K antagonist. DOAC: direct oral anticoagulation.

**Table 4 jcm-14-01173-t004:** Cox regression analysis showing predictors of composite outcomes.

	Univariate Analysis		Multivariate Analysis	
Presenting Characteristic	HR (95% CI)	*p*-Value	HR (95% CI)	*p*-Value
Female versus male	5.1 (3.2–8.1)	<0.001	6.2 (3.7–10.8)	<0.001
VKA versus DOAC	11 (5.7–20.9)	<0.001	8.4 (4.8–15.3)	<0.001
Age (≥60 years versus <60 years)	1.4 (0.7–2.7)	0.22		
Hypertension (yes versus no)	1.3 (0.6–3.8)	0.65		
Diabetes mellitus (yes versus no)	1.1 (0.58–4.9)	0.86		
Cerebrovascular disease (yes versus no)	1.3 (0.5–6.1)	0.47		
Ischaemic heart disease (yes versus no)	1.6 (0.6–3.6)	0.19		
CHA_2_DS_2_Vasc score (for one point increase)	1.4 (0.7–1.2)	0.09		
Previous percutaneous coronary intervention (yes versus no)	1.1 (0.6–1.5)	0.52		
Previous coronary artery bypass graft (yes versus no)	1 (0.9–1)	0.87		
Active thyroid disease (yes versus no)	1.1 (0.8–1.9)	0.76		
Chronic liver disease (yes versus no)	1.2 (0.4–2.1)	0.56		
Active malignancy (yes versus no)	1 (0.9–1)	0.4		
Chronic lung disease (yes versus no)	1 (0.9–1)	0.88		
Dementia (yes versus no)	1.1 (0.7–1.6)	0.78		
Anaemia (yes versus no)	1.1 (0.82–2.2)	0.83		
Previous bleeding event (yes versus no)	1.2 (0.4–1.8)	0.76		
Peripheral artery disease (yes versus no)	1.1 (0.8–1.4)	0.82		
On an antiplatelet on admission (yes versus no)	1.1 (0.3–2.1)	0.32		
On aspirin on admission (yes versus no)	1.2 (0.5–1.8)	0.25		
On clopidogrel on admission (yes versus no)	1.1 (0.9–1.4)	0.54		
On oral rate control medication (yes versus no)	0.8 (0.3–3.4)	0.2		
On oral rhythm control medication (yes versus no)	0.7 (0.2–2.8)	0.08		
Haemoglobin (every g/L increase)	0.8 (0.4–1.8)	0.61		
Total cholesterol (every mmol/L increase)	1.2 (0.2–3.1)	0.44		
Creatinine (every umol/L increase)	1.3 (0.6–2.4)	0.3		

VKA: vitamin K antagonist. DOAC: direct oral anticoagulation. HR: hazard ratio. CI: confidence interval. PCI: primary coronary intervention. CABG: coronary artery bypass graft.

## Data Availability

Data relating to this study are available upon reasonable request from the corresponding author.

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
