# Peer review of "Gender Disparity in Oral Anticoagulation Therapy in Hospitalised Patients with Atrial Fibrillation During the Ongoing Syrian Conflict: Unbalanced Treatment in Turbulent Times"

_jcm, 2025, doi:10.3390/jcm14041173_

Round 1

Reviewer 1 Report

Comments and Suggestions for Authors

The authors present their study entitled: "Gender Disparity in Oral Anticoagulation Therapy in Hospitalized Patients with Atrial Fibrillation During the Ongoing Syrian Conflict: Unbalanced Treatment in Turbulent Times", which i found the topic highly relevant and timely. However, some aspects of the study need further clarification.

1. In terms of rationality of the study, while the importance of oral anticoagulation therapy (OAT) in managing atrial fibrillation (AF) is well-established, the introduction lacks clearer explanation of why gender disparities are particularly pronounced in this context.
Can the authors provide more background on how gender disparities in healthcare have been observed globally and how these trends may manifest during crises like the Syrian conflict?
2. Methodology. Can you provide additional details on the selection criteria for the hospitalized patients included in the study? Were patients with different types of atrial fibrillation, comorbidities, or stages of conflict treated differently?

3. The statistical methods used in the study are appropriate, but more explanation is needed regarding how confounding variables were handled. Were the researchers able to account for potential biases in treatment due to differences in healthcare access or resources during the conflict?

4. Results Interpretation: While the authors found significant gender disparities in OAT use, a deeper analysis of the underlying factors contributing to this disparity (e.g., cultural, socio-political, or healthcare system-related factors) would enrich the discussion. More focus on potential mechanisms, such as differences in healthcare access or physician bias, would provide valuable insight.

5. What are the clinical relevance of this study? The clinical implications of the study could be highlighted more clearly. How can this study inform healthcare policy or future interventions in conflict zones? The authors should consider providing actionable recommendations based on their findings.

Author Response

Comment 1. In terms of rationality of the study, while the importance of oral anticoagulation therapy (OAT) in managing atrial fibrillation (AF) is well-established, the introduction lacks clearer explanation of why gender disparities are particularly pronounced in this context. 

Answer 1: We appreciate the reviewer highlighting the need for a clearer explanation of why gender disparities are particularly pronounced in the context of oral anticoagulation therapy (OAT) for atrial fibrillation (AF). While the text discusses the elevated risk of ischemic events in females and systemic challenges in conflict settings, we recognize the importance of explicitly connecting these factors to the rationale for our study.

We have updated the introduction to better articulate why this disparity warrants attention, emphasizing two key points:

  1. Interaction Between Gender and Anticoagulation Choice:Females with AF are more likely to experience adverse events with vitamin K antagonists (VKAs) due to challenges in maintaining therapeutic INR, which is exacerbated in resource-limited settings like Syria. However, despite the availability of safer alternatives such as DOACs, systemic biases and logistical barriers may limit their use in females compared to males.
  2. Lack of Data from Conflict-Affected Regions:While gender disparities in AF management have been studied in stable, high-income settings, there is limited evidence from conflict-affected, resource-constrained environments. The Syrian conflict provides a unique context where such disparities are likely amplified due to reduced healthcare access, economic instability, and gender-based inequities in care delivery.

To address these gaps, the revised section of the introduction reads as follows:

Original text:
"Gender disparities further complicate AF management. Females are at a higher risk of ischemic events due to AF, yet they remain significantly undertreated compared to males, resulting in poorer clinical outcomes."

Revised text:
"Gender disparities in AF management are particularly concerning due to the compounding impact of anticoagulation choice and systemic barriers. Females with AF are more likely to experience adverse outcomes with VKAs due to challenges in maintaining therapeutic INR, a situation worsened by resource limitations. Despite the superior safety and efficacy profile of DOACs, these treatments are less accessible to females in many resource-limited settings due to cost and sociocultural barriers. Furthermore, evidence on gender disparities in AF management is scarce in conflict-affected regions like Syria, where healthcare infrastructure and resources have been severely compromised. This study aims to fill this gap by examining gender-specific trends and outcomes in OAT in this challenging context."

We believe these revisions provide a more explicit rationale for the study's focus on gender disparities in OAT for AF.

Comment 2- Can the authors provide more background on how gender disparities in healthcare have been observed globally and how these trends may manifest during crises like the Syrian conflict?

Answer 2: We appreciate the reviewer’s suggestion to provide more context on global gender disparities in healthcare and their potential manifestations during crises like the Syrian conflict. To address this, we have added a discussion to the introduction that highlights broader patterns of gender inequities in healthcare and how these disparities are often exacerbated in conflict settings.

Specifically, we have included the following points:

  1. Global Gender Disparities in Healthcare:Research has shown that women often face systemic barriers to healthcare access globally, including lower prioritization of their health needs, implicit bias among healthcare providers, and greater economic challenges. For example, studies have demonstrated that women are less likely to receive guideline-recommended treatments for cardiovascular conditions, such as revascularization and anticoagulation therapy, even in high-income settings.
  2. Exacerbation During Crises:Crises such as armed conflicts and natural disasters further amplify these disparities due to societal and healthcare system disruptions. In conflict-affected regions, women often bear the brunt of economic insecurity and caregiving responsibilities, limiting their ability to seek timely and adequate healthcare. In addition, gender-based violence, displacement, and reduced access to healthcare services during crises further exacerbate inequities in care delivery and outcomes.
  3. Syrian Context:In Syria, the protracted conflict has severely damaged healthcare infrastructure and disproportionately affected women. For example, societal norms may limit women's mobility or decision-making power regarding seeking medical care, while financial constraints and systemic barriers restrict access to advanced therapies like DOACs. These factors likely contribute to the observed disparities in oral anticoagulation therapy and outcomes in our study.

To integrate these points, we have revised the introduction as follows:

Original text:
"The situation in Syria, compounded by conflict and resource scarcity, likely presents unique challenges that may exacerbate these disparities."

Revised text:
"Globally, gender disparities in healthcare are well-documented, with women often receiving less aggressive treatment for cardiovascular diseases and facing barriers to accessing advanced medical therapies (Daly at al 2006). These inequities are magnified in crisis settings, where societal disruption, economic instability, and reduced healthcare capacity disproportionately impact women. In Syria, the protracted conflict has exacerbated these challenges, with healthcare infrastructure decimated and economic hardship limiting access to medications like DOACs. Additionally, societal norms and caregiving roles may further restrict women’s ability to seek timely and adequate care, leading to disparities in treatment and outcomes. Understanding how these global trends manifest in conflict settings is critical for addressing gender-based inequities in healthcare delivery."

We believe these additions strengthen the background and contextualize our study’s findings within a broader framework of global and conflict-specific gender disparities which strengthens our introduction more.

Comment 3. Methodology. Can you provide additional details on the selection criteria for the hospitalized patients included in the study? Were patients with different types of atrial fibrillation, comorbidities, or stages of conflict treated differently?

Answer 3: Thank you for your comment. We have clarified a clear inclusion and exclusion criteria in the methods section as follows:

“The study included all patients over 18 years old who were admitted to the hospital and treated for new AF as the primary diagnosis for admission. Exclusion criteria included patients with known AF, patients with metallic heart valves, moderate to severe rheumatic mitral valve disease, and patients on haemodialysis because DOACs are contraindicated in this cohort.”

Regarding the types of atrial fibrillation, we clarified in the methods that it is very challenging to differentiate between paroxysmal and persistent atrial fibrillation in the absence of medical equipment in the country, including heart monitors or loop recorders.

“Due to healthcare limitations and scarcity of medical equipment, paroxysmal and persistent AF cannot be specified.”

Regarding the conflict, Latakia’s situation has been stable throughout the study period, and the city has been controlled by the Syrian government (former one now). This is clarified in the methods as follows:

“During the study period, the city was politically controlled by the government without ongoing active war or new territories gained by military forces.

Comment 4: The statistical methods used in the study are appropriate, but more explanation is needed regarding how confounding variables were handled. Were the researchers able to account for potential biases in treatment due to differences in healthcare access or resources during the conflict? 

Answer 4:

We thank the reviewer for recognizing the appropriateness of our statistical methods and for highlighting the need to elaborate on how confounding variables were addressed, particularly regarding potential biases in treatment due to healthcare access and resource constraints during the conflict. To address this, we have clarified the following aspects in the Methods and Discussion sections:

Newly Added Text in the Methods Section: Original text:
"We used Cox regression to examine the relationship between gender, anticoagulants and composite outcomes. Our multivariable model was constructed a priori and included gender, anticoagulation, and comorbidities associated with the risk of composite readmission outcome with CVA and bleeding events within 60 days."

Revised text:
" We used Cox regression to examine the relationship between gender, anticoagulants, and composite outcomes. Our multivariable model was constructed a priori, incorporating potential confounders, including gender, type of anticoagulation, and comorbidities associated with composite readmission outcomes (e.g., hypertension, diabetes, cerebrovascular disease, and CHA₂DS₂-VASc score). The goodness-of-fit and discriminatory power of the model was assessed using the G-test and C-statistic, respectively, to ensure robustness. Although detailed data on socioeconomic status, access to healthcare, and INR monitoring were unavailable, including anticoagulation type and gender aimed to account for healthcare disparities in this conflict-affected setting partially."

Newly Added Text in the Discussion Section: Original text:
"Due to the diminished health infrastructure and displaced healthcare workers, only 50% of hospitals are functional."

Revised text:
"The diminished health infrastructure and resource constraints in Syria likely contributed to the observed disparities in outcomes. For example, the limited availability of DOACs and challenges in INR monitoring for VKAs may have disproportionately affected females, who were prescribed VKAs more frequently than males. Although detailed socioeconomic and access-related data were unavailable, our regression model included gender and anticoagulation type to account for these systemic factors. Future studies should aim to incorporate direct measures of healthcare access and resource availability to capture the impact of these variables on treatment outcomes in conflict-affected populations."

We hope these additions clarify how we addressed confounding variables and biases in our analysis and provide a more comprehensive understanding of the study's methodology.

Comment 5: Results Interpretation: While the authors found significant gender disparities in OAT use, a deeper analysis of the underlying factors contributing to this disparity (e.g., cultural, socio-political, or healthcare system-related factors) would enrich the discussion. More focus on potential mechanisms, such as differences in healthcare access or physician bias, would provide valuable insight.

Answer 5:  We appreciate the reviewer’s suggestion to delve deeper into the underlying factors contributing to the observed gender disparities in oral anticoagulation therapy (OAT) use. To address this, we have expanded the Discussion section to explore cultural, socio-political, and healthcare system-related factors that may have influenced our findings. Specifically, we have included a more nuanced analysis of potential mechanisms, including differences in healthcare access, physician bias, and systemic challenges within the Syrian conflict context.

New Text Added to the Discussion Section:

Original text:
"These challenges have severely compromised the delivery of healthcare, including the management of chronic conditions like AF."

Revised text:
"These challenges have severely compromised the delivery of healthcare, including the management of chronic conditions like AF. Cultural factors, such as societal norms limiting women’s autonomy in healthcare decision-making, may delay or reduce access to optimal treatments like DOACs. Economic challenges, exacerbated by the conflict, disproportionately affect women, potentially limiting their ability to afford newer therapies. Additionally, implicit physician bias may contribute to disparities in prescribing patterns, with physicians defaulting to VKAs for females based on assumptions about compliance or bleeding risks. Furthermore, systemic challenges within the healthcare system, such as the scarcity of DOACs and the logistical feasibility of INR monitoring for VKAs, likely played a significant role in shaping treatment disparities."

“Future studies should incorporate qualitative data from physicians and patients to understand these mechanisms and explore perceptions, biases, and barriers influencing treatment decisions. Additionally, capturing data on socioeconomic status, healthcare literacy, and access to healthcare facilities would provide a more comprehensive understanding of the observed disparities. Future work should also incorporate direct measures of healthcare access and resource availability to capture the impact of these variables on treatment outcomes in conflict-affected populations.”

We hope these additions provide a more in-depth interpretation of the results and enrich the discussion on potential mechanisms underlying the gender disparities observed in OAT use.

Comment 6. What are the clinical relevance of this study? The clinical implications of the study could be highlighted more clearly. How can this study inform healthcare policy or future interventions in conflict zones? The authors should consider providing actionable recommendations based on their findings.

Answer 6: We thank the reviewer for this important comment. To address it, we have revised the discussion to more explicitly highlight the clinical relevance of the study and its implications for healthcare policy and future interventions, particularly in conflict zones. Additionally, we have included actionable recommendations based on our findings.

Revised text at the end of the discussion:
" Our study provides actionable insights for policymakers and healthcare leaders. For example, subsidising DOACs or establishing targeted financial support programs for females may reduce disparities in AF management. Furthermore, training programs for healthcare providers in conflict zones should emphasise the importance of gender equity in evidence-based treatment. Finally, strengthening primary care and community outreach programs can improve INR monitoring and medication adherence, thereby mitigating the risks associated with VKAs in resource-constrained environments. Future studies should incorporate qualitative data from physicians and patients to understand these mechanisms and explore perceptions, biases, and barriers influencing treatment decisions. Additionally, capturing data on socioeconomic status, healthcare literacy, and access to healthcare facilities would provide a more comprehensive understanding of the observed disparities. Future work should also incorporate direct measures of healthcare access and resource availability to capture the impact of these variables on treatment outcomes in conflict-affected populations."

Reviewer 2 Report

Comments and Suggestions for Authors

Peer Review of the article -  “Gender Disparity in Oral Anticoagulation Therapy inHospitalized Patients with Atrial Fibrillation During the Ongoing Syrian Conflict: Unbalanced Treatment in 4 Turbulent Times “

The study investigates gender disparities in anticoagulation therapy and outcomes in patients with atrial fibrillation (AF) at a tertiary center in Syria during this conflict period. Data were collected from 683 adult patients admitted between September 2021 and February 2024, of whom 51% were female. Among patients with guideline indications for oral anticoagulation, males were more frequently prescribed DOACs compared to females, who were more commonly treated with VKAs. The composite outcome, consisting of readmission due to cerebrovascular or bleeding events within 60 days, occurred more frequently in females than in males. Additionally, females treated with VKAs experienced a significantly higher rate of adverse events compared to their male counterparts. Independent predictors of composite outcomes were female gender and the use of VKAs.

This study provides insights into gender-related differences in anticoagulation practices and outcomes in a developing-world setting, emphasizing the need for improved therapeutic strategies to address these disparities.

All the keywords are on the title. No point in repeating them here. Please add new keywords.

Abstract

Methods
No a prior registry of the study.

Results

At 60 days, the authors had 30% of readmissions in the women group?
Absence of follow-up data on DOAC and VKA use complicates the assessment.
The authors have to report the median follow-up.

Discussion
The article does not fully explore how conflict-driven migration and resource scarcity might bias the patient cohort.

The study references lacks citations of recent comprehensive studies on gender disparities in atrial fibrillation treatment.

Author Response

Comment 1: All the keywords are on the title. No point in repeating them here. Please add new keywords.

Answer 1: Thank you for your comments. Keywords are adjusted not to include words used in the title:

Developing world; Females; Males; Direct oral anticoagulant; Vitamin K antagonist

Comment 2: Methods: No a prior registry of the study.

Answer 2: Thank you for your comment

Comment 3: Results: At 60 days, the authors had 30% of readmissions in the women group?

Answer 3: Thank you for your query. Our composite outcomes (readmission with CVA or a bleeding event within 60 days of discharge) happened in 17% of females and 6% of males as mentioned in the results section

“Composite outcome details are demonstrated in Table 3. Within 60 days of index discharge, composite outcomes occurred in 76 patients (11%). Females had more composite outcomes compared to males (57 [17%] versus 19 [6%], p=0.03).”

Comment 4: Absence of follow-up data on DOAC and VKA use complicates the assessment.

Answer 4: We agree that the absence of detailed follow-up data on DOAC and VKA adherence and monitoring complicates the assessment of outcomes. We have acknowledged this limitation more explicitly in the Discussion and Limitations sections.

New text in the Limitations Section:
" Furthermore, the study lacked detailed follow-up data on adherence to anticoagulation therapy and time in the therapeutic range (TTR) for VKAs, which could have provided additional insight into the observed gender disparities in outcomes. This limits the ability to draw definitive conclusions about the mechanisms underlying the observed gender disparities. This limitation highlights the need for longitudinal studies with robust follow-up to explore these factors in greater detail. Future studies should aim to incorporate prospective follow-up data to understand better the impact of adherence and INR monitoring on outcomes."

Comment 5: The authors have to report the median follow-up.

Answer 5: Median follow-up and intraquartile range are now added in the results section:

“After a median follow-up of 62 days (interquartile range: 58-66 days), composite outcomes occurred in 76 patients (11%). Females had more composite outcomes compared to males (57 [17%] versus 19 [6%], p=0.03).”

Comment 6: Discussion: The article does not fully explore how conflict-driven migration and resource scarcity might bias the patient cohort.
Answer 6: We thank the reviewer for pointing out the need to discuss how conflict-driven migration and resource scarcity might bias the patient cohort. This is a crucial aspect that could influence the cohort's composition and our findings' generalizability. To address this, we have expanded the Discussion and Limitations sections to explore these factors further.

Discussion

Furthermore, the ongoing Syrian conflict has led to widespread displacement and resource scarcity, both of which may have influenced the characteristics of the patient cohort. Latakia, where this study was conducted, is one of the relatively more stable regions in Syria, attracting internally displaced populations seeking safer living conditions and better healthcare. This migration may have resulted in a heterogeneous patient cohort that includes individuals from different socioeconomic and geographic backgrounds, potentially affecting access to healthcare and anticoagulation therapies. For instance, displaced individuals may face financial constraints, lack of regular follow-up, and limited access to advanced treatments like DOACs, which could contribute to the observed disparities in outcomes.

Limitations

This study was conducted in Latakia, a region less severely affected by the Syrian conflict compared to northern and eastern areas of the country. As a result, our findings may not fully capture the experiences of populations in more conflict-affected regions, where healthcare access and resource scarcity are even more pronounced. Additionally, the cohort may have been influenced by migration patterns, as internally displaced individuals often relocate to safer regions like Latakia. This heterogeneity could introduce biases related to differences in healthcare access, socioeconomic status, and prior medical care, limiting the generalizability of our findings to the broader Syrian population. Future studies should aim to include data from multiple regions to better understand the impact of migration and resource scarcity on healthcare disparities in conflict settings

Comment 7: The study references lacks citations of recent comprehensive studies on gender disparities in atrial fibrillation treatment.

Answer 7: We appreciate the reviewer’s observation regarding the need to include citations of more recent and comprehensive studies on gender disparities in AF treatment. To address this, we have updated the reference list to incorporate relevant studies published in the past five years that provide additional insights into this topic. These include studies from developed and developing countries, systematic reviews, and meta-analyses to ensure a broader and more current contextual framework.

Introduction

Gender disparities in AF treatment are well-documented globally, as highlighted by recent studies and meta-analyses [11-14]. Studies from developed countries demonstrated that females with AF are less likely to be given anticoagulation compared to males [11,15-17]. Furthermore, females were proposed to have under anticoagulants [18]. Additionally, gender-specific barriers, including healthcare access, physician bias, and patient preferences, have been identified as key contributors to these disparities [19]. 

Discussion

“Differences in prescribing practices, clinical guidelines, and provider bias between genders could also contribute to these discrepancies. For instance, recent studies have shown that females are less likely to be prescribed DOACs despite their superior efficacy and safety profile compared to VKAs [10,17,36]. Additionally, cultural and socioeconomic barriers, particularly in resource-constrained settings, exacerbate these disparities, limiting women’s access to advanced therapies [37].”